# Effect of Different Warm-Up Durations on the Plasma Oxidative Stress Biomarkers Following Anaerobic Exercise in Amateur Handball Players

**Abdessalem Koubaa [1,2], Sirine Koubaa [1] and Mohamed Elloumi [3,*]**

[1] High Institute of Sport and Physical Education, University of Sfax, Sfax 3000, Tunisia; abdessalemkoubaa@gmail.com (A.K.); sirinekoubaa6@gmail.com (S.K.)
[2] Laboratory of Pharmacology, Faculty of Medicine, University of Sfax, Sfax 3029, Tunisia
[3] Sport Sciences and Diagnostics Research Group, GS-HPE Department, Prince Sultan University, Riyadh 11586, Saudi Arabia
[*] Correspondence: melloumi@psu.edu.sa

**Abstract:** The aim of this study was to assess the effects of three warm-up durations on post-exercise oxidative stress biomarkers, as well as recovery from the Wingate test in 14 amateur handball players. These players completed the Wingate test three times: after 5 min, after 10 min, and then after 15 min of warm-up, spread over 2 weeks with an interval of 1 week between each session. At the end of each session of the Wingate test, blood samples were taken: at rest (PR), after warming up (PWU) and immediately at the end of the test (P0), then after 10-min (P10) and after 20-min (P20) of recovery. The measured parameters are the total antioxidant status (TAS), superoxide dismutase (SOD), glutathione reductase (GR), glutathione peroxidase (GPx), malondialdehyde (MDA), protein-bound-carbonyls (PCs) and thiobarbituric acid reactive substances (TBARSs). The main findings revealed that anaerobic exercise induces oxidative stress, as evidenced by changes in antioxidant status and significant increases in concentrations of the majority of oxidative stress indicators ($p < 0.05$). At P20, plasma GPx, SOD, GR, TBARSs, and PC contents, are lower and significantly different after a 15-min warm-up than after a 10-min or 5-min warm-up. ANOVA showed a significant "warm-up × recovery" effect on plasma GPx, SOD, GR, TBARSs, and PC contents [$F(8.104) = 3.61$; $p < 0.001$; $\eta^2 p = 0.22$, $F(8.104)$: from 1.88 to 19.97; $\eta^2 p$: from 0.19 to 0.61: $0.05 < p < 0.001$], respectively. We concluded that a 15-min warm-up was more efficient than the other duration in reducing a free radical rise, emphasizing the importance of the warm-up length on plasma oxidative stress biomarkers.

**Keywords:** warm-up; oxidative stress; anaerobic exercise; handball players

## 1. Introduction

Muscle damage is one of the most common issues that today's athletes face, whether they are amateurs or professionals [1]. Indeed, skeletal muscle injuries account for more than 30% of injuries evident in sports medicine consultations, according to Kirkendall DT et al. [2]. As a result, it is essential to employ the most effective methods to prevent these injuries. Several prior studies have suggested that warming up may play an important role in reducing muscle damage, whereas other investigations have found no positive effects. In fact, the significance of stretching and warming up in reducing musculoskeletal injuries has been debated. Some studies found no reduction in injury risk or the total number of injuries [3], whereas others found positive effects and reduced neuromuscular injuries after stretching or warming up programs [4]. Therefore, a number of investigations has found that the justifications supporting the preventive function of warming up and stretching on muscle damage are insufficient [5,6]. Several studies have found a link between an increase in free radicals and hence oxidative stress and the development of muscle injury [7].

Furthermore, studies on the effects of anaerobic exercise on oxidative stress are few in comparison to studies on the effects of aerobic exercise [8,9]. These studies, however, reveal an increase in oxidative stress following supramaximal exercise. To defend against the potentially harmful effects of oxidative stress, organisms have anti-free radical defense systems that involve a variety of enzymatic antioxidants (superoxide dismutase (SOD); glutathione peroxidase (GPX); catalase and non-enzymatic antioxidants (Vitamin E and C; zinc; selenium; copper; uric acid) [10]. In addition to the role of the body's anti-free radical system, many strategies for limiting and modalities for preventing the increase in pro-oxidants have been proposed in the sports field, such as variation in duration, intensity and type of exercise [11], and the type of recovery [12,13]. Recent research has focused on proving an increase in total antioxidant status after anaerobic exercise in amateur athletes [14].

Warming up helps prepare the muscles for exercise by increasing their contractile capacity [15] and decreasing muscle and tendon viscoelasticity [16]. Warming up has been shown to increase the energy supply to the muscles, increase the rate of nerve impulse transmission, reduce internal viscosity, and increase muscle activation, strength, blood supply, and oxygen to the working muscle [17–19]. The effects of warming up on cardiorespiratory functions [20], the psychological state [21], and the risk of injury [22] have been well documented. However, its impact on oxidative stress indicators has received little attention. In this regard, we propose to conduct a study to evaluate the potential short-term influence of varied warm-up durations on muscle injury prevention and the evolution of oxidative stress biomarkers after anaerobic exercise in young amateur handball players. Therefore, we hypothesize that a 15-min warm-up could regulate and favorably attenuate the negative effects of oxidative stress, potentially reducing muscle damage after anaerobic exercise.

## 2. Materials and Methods

### 2.1. Participants

A total of 14 male handball players belonging to the regional selection of Sfax-Tunisia was recruited to participate in this study. They have been routinely training for at least 8 years ($8.7 \pm 0.3$ years) at a rate of five sessions per week. The anthropometric characteristics of all participants are shown in Table 1.

**Table 1.** The anthropometric characteristics of the participants (Means $\pm$ SD).

| Parameters | Means $\pm$ SD |
|:---:|:---:|
| Age (yrs) | $22.6 \pm 0.3$ |
| Weight (kg) | $74.2 \pm 0.5$ |
| Height (cm) | $183 \pm 6.4$ |
| BMI (kg.m$^{-2}$) | $24.6 \pm 1.4$ |
| FFM kg) | $59.4 \pm 1.3$ |
| LBM (kg) | $63.8 \pm 3$ |

Note: BMI, body mass index; FFM, fat free mass; LBM, lean body mass.

After receiving a verbal description of the protocol, study risks and benefits, all players gave their written consent before participating in our experimental protocol, which was approved by the Research Ethics Committee of the Faculty of Medicine, University of Sfax, Tunisia (protocol number IRB00009853-03). None of the participants was injured or ill, they were all non-smokers, had no pathological sleep disorders, did not consume alcohol, were not regular creatine users, and had not used antioxidant and polyphenol supplementation within the 3 months preceding the study, and they were all regularly training.

### 2.2. Experimental Design

According to the results of Zois et al. [23], warm-up routines lasting more than 5 to 15 min may have a negative impact on subsequent performance due to excessive fatigue.

Similarly, Pardeiro and Yanci [24] reported that a 25-min warm-up reduced physical performance in semi-professional soccer players. As a result, we set these three warm-up intervals (5, 10, and 15 min) to avoid fatigue, which may impair physical performance.

In order to reduce the influence of exogenous factors on biomarkers of oxidative stress, participants were asked to stop eating at least 2 to 3 hours before the experiment, not to take alcohol, caffeine, or any other energizing drug, and not to engage in strenuous physical activities during the 24 h prior to the experiment. Participants performed three sessions, in the first phase of the transition period, in random order, with a week of rest between each session. The first session begins with a 5-min pedal warm-up, followed by a 30-s Wingate test on an ergocycle (Monark type 894 E) with an imposed load of 75 g/kg of body mass. The second session, which took place a week later, consisted of a 10-min pedaling warm-up followed by a 30-s Wingate test. The third session, which took place a week later, consisted of a 15-min cycling warm-up followed by a 30-s Wingate test.

Prior to the start of the experiment, the subjects performed an incremental peak test on the ergometer to determine the intensity of the warm-up. To do this, the participant pedaled for 3 min at 50 W, then an increase in the intensity of 25 W every 2 min until exhaustion.

The warm-up intensity was chosen on the basis of the output power recorded in the last step of the incremental test (50% of this power). To neutralize the chrono-biological effects, all sessions were carried out at 9 a.m.

### 2.3. Anthropometric Measurements and Body Composition

Respecting the criteria of the American College of Sports Medicine (ACSM) [25], body mass was measured with an electronic scale (TANITATBF.350), and standing height was measured with a measuring rod. BMI was calculated using the formula: BMI = mass (kg)/height$^2$ (m$^2$). Fat free mass (FFM) and lean body mass (LBM) were calculated according to the Janmahasatian [26] and Yu [27] equations, respectively:

$$FFM = (9270 \times weight)/(6680 + (216 \times BMI)$$

$$LBM = 22.93 + 0.68 \ (weight) - 1.14 \ (BMI) - 0.01 \ (age) + 9.94$$

### 2.4. Dietary Program

We applied standardized caloric and hydric intake for the 3 days preceding each testing that could alter oxidative stress in this investigation. A daily dietary record was kept for 3 days to examine the sufficiency and consistency of nutrient consumption. Bilnu 4 software (SCDA Nutrisoft, Cerelles, France) and food composition tables issued by the Tunisian National Institute of Statistics in 1978 were used to assess the individuals' diets. The estimated nutrient intakes were compared to the reference dietary intakes for physically active people, and the daily nutrient data revealed that total calorie, macronutrient, and micronutrient intakes were all within the reference dietary intakes for healthy Tunisian adults, with no significant differences between the three test sessions (Table 2).

**Table 2.** Daily calorie intake and percentages of carbohydrates, fats and proteins recorded during the different sessions (mean ± SD).

| Variables | Mean (±SD) | | |
|---|---|---|---|
| | Session 1 | Session 2 | Session 3 |
| Calorie intake (kcal/day) | 3153 ± 277 | 3271 ± 485 | 3244 ± 410 |
| Carbohydrate (%) | 53.4 ± 5.1 | 51.7 ± 5.4 | 52.3 ± 5.2 |
| Lipids (%) | 33.7 ± 5.6 | 35.1 ± 2.3 | 34.5 ± 4.5 |
| Protein (%) | 12.5 ± 1.2 | 11.9 ± 2.1 | 12.3 ± 1.7 |

## 2.5. Blood Sampling and Biochemistry

The experimental protocol consists of taking several samples: At rest (PR), immediately after the warm-up (PWU), immediately after the Wingate test (P0), and within 10 min and 20 min of stopping the exercise, respectively (P10) and (P20) (Figure 1). The experiments and analyses were carried out in the Pharmacology Laboratory of the Faculty of Medicine in Sfax, Tunisia. The conditions for blood samples closely adhere to ethical guidelines. Medical biologists used an intravenous catheter to collect these samples from the pharmacology laboratory in order to examine the evolution of plasma antioxidants. After 10 h of fasting and 9 h of sleep, blood samples were collected at 8 a.m. under basal conditions. They were drawn from an antecubital vein, centrifuged at 4000 rpm to produce plasma, and then kept in aliquots at −80 °C until analysis.

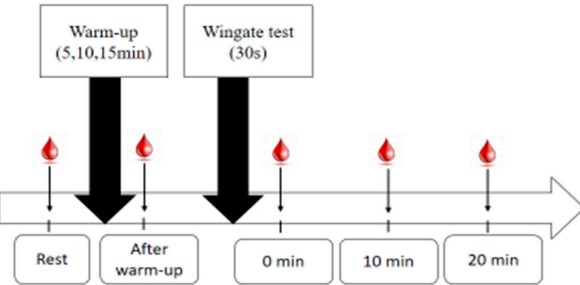

**Figure 1.** Study design.

Analysis of oxidative stress biomarkers was performed using dosage kits (Randox Laboratories, place-country-region UK). Standard colorimetric dosages were used to measure plasma concentrations of glutathione reductase (GR), glutathione peroxidase (GPx), superoxide dismutase (SOD), and total antioxidant status (TAS). Malondialdehyde (MDA) concentrations were measured by the formula: Sample = (sample peak height × calibrator concentration)/peak height calibrator, and analyzed via a HPLC malondialdehyde procedure.

Protein carbonyls after their derivatization with 2,4 dinitrophenylhydrazine (DNPH) were detected by spectrophotometry at 370 nm, referring to the basic methods described by Levine et al. [28]. Using tetramethoxypropane as a standard, plasma TBARSs were measured using the method of Yagi [29].

## 2.6. Statistical Analyses

All statistical tests were processed using STATISTICA Software 20 (Stat-Soft, Paris, France). Following normality, confirmation was assessed using the Shapiro–Wilk's W-test, and analysis of the results for each parameter was performed using a two-factor analysis of variance (ANOVA). These ANOVAs took the following form: 3 "warm-up durations" (5-min warm-up; 10-min warm-up and 15-min warm-up) × 5 "blood samples" (PR; PWU; P0; P10; P20) with repeated measurements on both factors. When ANOVA showed significant effects, Bonferroni-type post hoc tests were applied to make two-by-two comparisons of the experimental data. All the observed differences are considered to be statistically significant for a probability threshold less than 0.05 ($p < 0.05$). The required sample size was calculated using G*power software [30]. The values for $\alpha$ and power were set to 0.05 and 0.80, respectively. Effect sizes were calculated using partial eta-squared ($\eta P_2$) and the magnitudes were interpreted using the thresholds: <0.2 was considered small, around 0.5 was considered medium and >0.8 was considered large [31]. The effect size was estimated to be 0.6 (medium effect) based on the study of El Abed et al. [32]. The required sample size was 14.

## 3. Results

The differences in concentrations of plasma oxidative stress biomarkers for the three warm-up times are summarized in Table 3.

**Table 3.** Concentrations (Means ± SD) of oxidative stress markers as a function of recovery and duration of warm-up.

| Variables | | Means ± Standard Deviations | | | | | Warm-Up | | | Recovery | | | Warm-Up × Recovery | | |
|---|---|---|---|---|---|---|---|---|---|---|---|---|---|---|---|
| | | Rest | PWU | P0 | P10 | P20 | F(2, 26) | $p$ | $\eta^2$p | F(4, 52) | $p$ | $\eta^2$p | F(8, 104) | $p$ | $\eta^2$p |
| SAT (mmol/L) | Ech15 | 1.51 ± 0.05 | 1.53 ± 0.05 | 1.75 ± 0.05 ab | 1.79 ± 0.04 ab | 1.8 ± 0.04 ab | 10.17 | <0.001 | 0.44 | 246.96 | <0.001 | 0.95 | 0.87 | 0.544 | 0.06 |
| | Ech10 | 1.53 ± 0.04 | 1.57 ± 0.03 | 1.77 ± 0.05 ab | 1.8 ± 0.05 ab | 1.82 ± 0.04 ab | | | | | | | | | |
| | Ech05 | 1.54 ± 0.09 | 1.58 ± 0.05 | 1.79 ± 0.07 ab | 1.83 ± 0.07 ab | 1.87 ± 0.07 abc* | | | | | | | | | |
| SOD (U/g Hg) | Ech15 | 1316 ± 49 | 1363 ± 41 | 1528 ± 70 a | 1491 ± 70 | 1386 ± 60 | 27.75 | <0.001 | 0.68 | 18.49 | <0.001 | 0.59 | 2.16 | 0.037 | 0.14 |
| | Ech10 | 1375 ± 131 | 1468 ± 177 | 1593 ± 177 a | 1672 ± 198 ab | 1687 ± 210 ab* | | | | | | | | | |
| | Ech05 | 1409 ± 151 | 1456 ± 181 | 1557 ± 192 | 1601 ± 155 | 1662 ± 173 ab* | | | | | | | | | |
| GPx (U/g Hg) | Ech15 | 35.2 ± 1.5 | 37.01 ± 1.71 | 40.32 ± 2.32 a | 39.22 ± 1.67 a | 36.97 ± 1.62 | 1.28 | 0.295 | 0.09 | 59.52 | <0.001 | 0.82 | 3.61 | <0.001 | 0.22 |
| | Ech10 | 34.81 ± 3.84 | 35.55 ± 3.95 | 41.19 ± 4.49 ab | 42.4 ± 4.3 ab | 41.55 ± 2.59 ab* | | | | | | | | | |
| | Ech05 | 34 ± 3.88 | 35.24 ± 3.78 | 39.73 ± 5.24 ab | 40.6 ± 5.36 ab | 40.87 ± 3.75 ab* | | | | | | | | | |
| GR (U/g Hg) | Ech15 | 9.64 ± 0.57 | 10.08 ± 0.52 | 12.17 ± 0.59 | 10.78 ± 0.45 | 9.76 ± 0.5 | 5.65 | 0.009 | 0.30 | 13.13 | <0.001 | 0.50 | 1.88 | 0.071 | 0.13 |
| | Ech10 | 9.43 ± 0.59 | 9.94 ± 0.56 | 12.73 ± 0.52 | 12.6 ± 0.82 | 12.21 ± 0.94 | | | | | | | | | |
| | Ech05 | 9.56 ± 0.49 | 9.99 ± 0.48 | 13.38 ± 0.29 ab | 13.61 ± 0.31 ab | 13.67 ± 9.76 ab* | | | | | | | | | |
| MDA (μmol/L) | Ech15 | 1.56 ± 0.05 | 1.59 ± 0.06 | 1.8 ± 0.07 ab | 1.85 ± 0.03 ab | 1.85 ± 0.04 ab | 0.31 | 0.737 | 0.02 | 83.95 | <0.001 | 0.87 | 1.01 | 0.435 | 0.07 |
| | Ech10 | 1.57 ± 0.08 | 1.62 ± 0.06 | 1.8 ± 0.07 ab | 1.82 ± 0.05 ab | 1.8 ± 0.05 ab | | | | | | | | | |
| | Ech05 | 1.61 ± 0.23 | 1.65 ± 0.22 | 1.81 ± 0.09 ab | 1.83 ± 0.08 ab | 1.82 ± 0.05 ab | | | | | | | | | |
| TBARS (mmol/L) | Ech15 | 0.29 ± 0.05 | 0.32 ± 0.06 | 0.37 ± 0.05 ab | 0.35 ± 0.05 a | 0.31 ± 0.05 cd | 3.59 | 0.042 | 0.22 | 161.52 | <0.001 | 0.93 | 12.24 | <0.001 | 0.48 |
| | Ech10 | 0.28 ± 0.06 | 0.31 ± 0.05 | 0.4 ± 0.07 ab | 0.4 ± 0.05 ab* | 0.4 ± 0.03 ab* | | | | | | | | | |
| | Ech05 | 0.28 ± 0.05 | 0.32 ± 0.05 | 0.39 ± 0.04 ab | 0.43 ± 0.02 ab* | 0.42 ± 0.02 ab* | | | | | | | | | |
| PC (nmol/mg protein) | Ech15 | 0.48 ± 0.07 | 0.52 ± 0.07 | 0.64 ± 0.07 ab | 0.6 ± 0.07 ab | 0.5 ± 0.06 cd | 13.28 | <0.001 | 0.51 | 306.36 | <0.001 | 0.96 | 19.97 | <0.001 | 0.61 |
| | Ech10 | 0.49 ± 0.06 | 0.53 ± 0.06 | 0.71 ± 0.04 ab* | 0.69 ± 0.03 ab* | 0.67 ± 0.05 ab* | | | | | | | | | |
| | Ech05 | 0.49 ± 0.07 | 0.54 ± 0.07 | 0.72 ± 0.04 ab* | 0.74 ± 0.03 ab* | 0.73 ± 0.03 ab*# | | | | | | | | | |

Note: a: significant difference compared to rest at $p < 0.05$; b: significant difference compared to PWU at $p < 0.05$; c: significant difference compared to P0 at $p < 0.05$; d: significant difference compared to P10 at $p < 0.05$; *: significant difference compared to warm-up 15 min at $p < 0.05$; #: significant difference compared to warm-up 10-min at $p < 0.05$.

Concerning the TAS concentrations, the ANOVA showed a significant "warm-up" effect ($F_{(2, 26)} = 10.17$; $p < 0.01$; $\eta^2 p = 0.44$), and a significant "recovery" effect ($F_{(4, 52)} = 246.96$; $p < 0.001$; $\eta^2 p = 0.95$). Statistical analysis showed a significant difference ($p < 0.05$) between PR and P0 for the three warm-up durations performed and a significant difference between PWU and P0 for the three warm-up durations performed.

Following a 15-min warm-up and after 20 min of effort, the SAT concentrations are significantly different ($p < 0.05$) from those found after 5 min of warm-up (Figure 2).

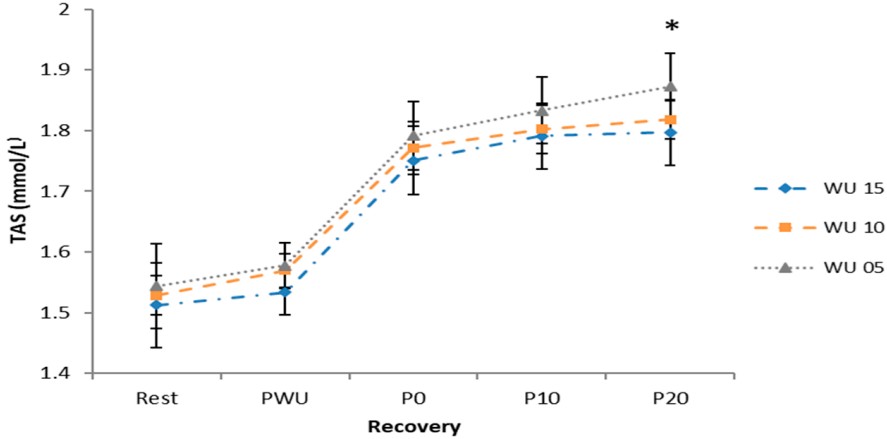

**Figure 2.** Evolution of the TAS at recovery after the three warm-up durations. *: significant difference compared to 15-min warm-up at $p < 0.05$.

The 5-min, 10-min or 15-min warm-up durations induced variable changes in plasma SOD concentrations during the recovery. Figure 3 shows that 20 min after exercise following a 15-min warm-up, SOD concentrations are lower than and significantly different from those found after a 10- or 5-min warm-up (Figure 3). Statistical analysis revealed that SOD concentrations increased immediately after the Wingate test ($p < 0.05$).

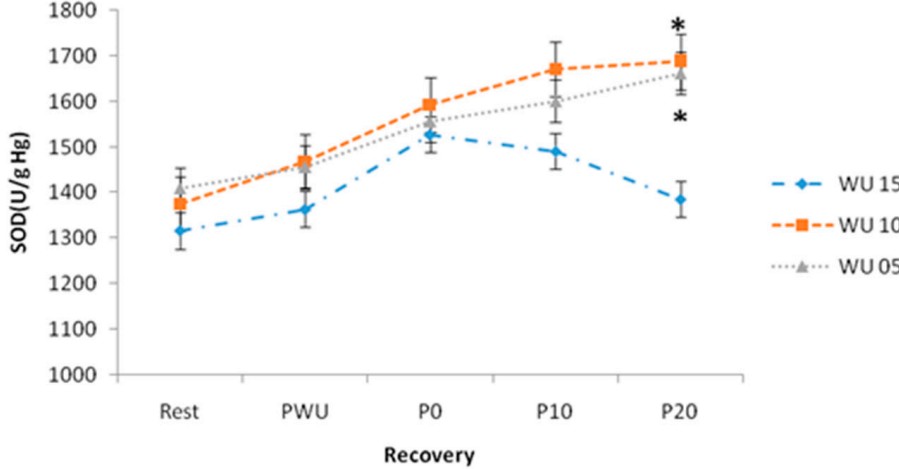

**Figure 3.** Evolution of SOD at recovery after the three warm-up durations. *: significant difference compared to 15-min warm-up at $p < 0.05$.

Concerning GPx concentrations, ANOVA showed a significant "recovery" effect ($F_{(4, 52)} = 59.52$; $p < 0.001$; $\eta^2 p = 0.82$) and a significant "warm up × recovery" interaction ($F_{(8, 104)} = 3.61$; $p < 0.001$; $\eta^2 p = 0.22$) (Figure 4).

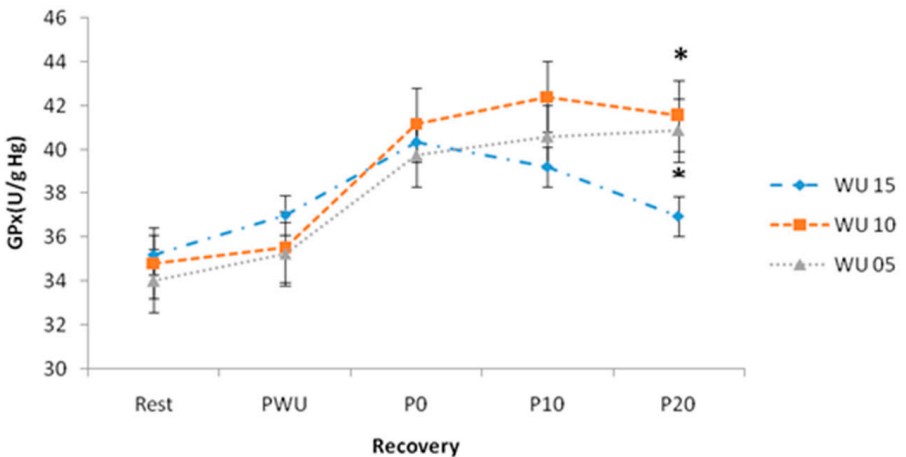

**Figure 4.** Evolution of GPX at recovery after the three warm-up durations. *: significant difference compared to15-min warm-up at *p* < 0.05.

In addition, a significant difference was recorded between the concentrations of GPx at rest and the concentrations measured after exercise for the 15-min warm-up period.

After 20 min of recovery, the GPx concentrations are higher following a warm up of 10-min and 5 min compared to those recorded after a 15-min warm-up (*p* < 0.05; Table 2).

With regard to the concentrations of the GR, the ANOVA showed a significant "warm up" effect (F (2, 26) = 5.65; *p* < 0.01; η²p = 0.30), a significant "recovery" effect (F (4, 52) = 13.13; *p* < 0.001; η²p = 0.50) and an insignificant "warm up × recovery" interaction (F (8, 104) = 1.88; *p* > 0.05; η²p = 0.13). Statistical analysis showed a significant difference (*p* < 0.05) of GR at P0 compared to PR and PWU for the 5-min warm-up time (Table 2). Furthermore, a significant difference was recorded in the plasma GR contents within 20 min of stopping the exercise test between the two warm-up times (5 min and 10 min), the values GR of which are, respectively, 13.67 ± 9.76 (U/g Hg) and 9.76 ± 0.5 (U/g Hg) (Figure 5).

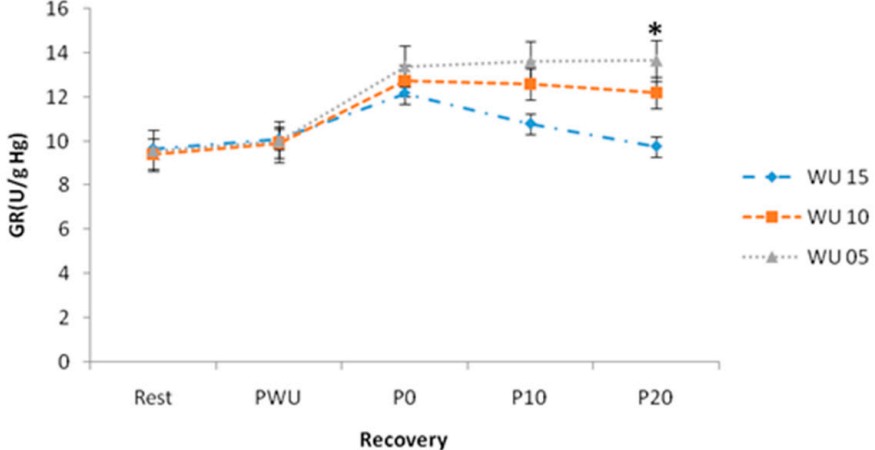

**Figure 5.** Evolution of GR at recovery after the three warm-up durations. *: significant difference compared to15-min warm-up at *p* < 0.05.

The three warm-up times induced significant increases in MDA concentrations upon stopping exercise (P0) at P10 and P20 within up to 20 min of recovery. By comparing MDA concentrations during recovery versus P resting and PWU, ANOVA showed significant differences for the different warm-up times (Table 1). The statistical analysis noted that following the three warm-up times, the MDA concentrations increased, from the stop of the Wingate test until 20 min of recovery without significant differences recorded to reach

1.85 ± 0.04 (µmol/L), 1.8 ± 0.05 (µmol/L) and 1.82 ± 0.05 (µmol/L) following 15 min of warm-up; 10 min and 5 min respectively (Figure 6).

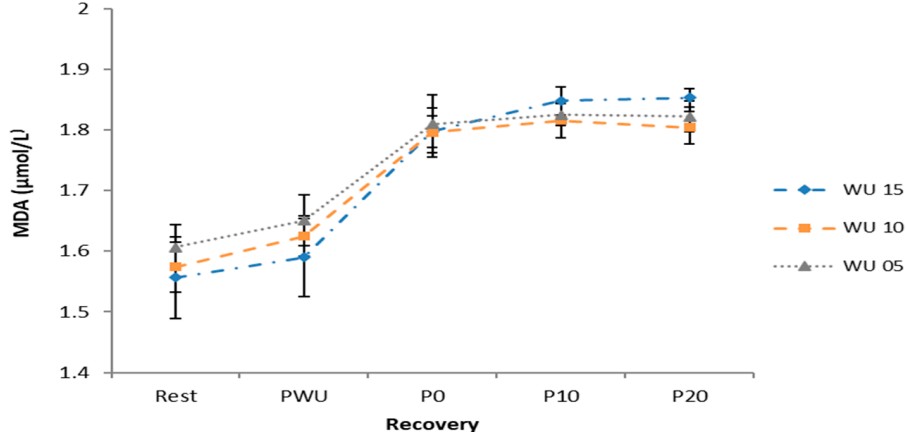

**Figure 6.** Evolution of the MDA at recovery after the three warm-up durations.

By examining TBARSs evolution, ANOVA showed a significant "warm-up" effect (F (2, 26) = 3.59; $p < 0.05$; $\eta^2 p = 0.22$), a significant "recovery" effect (F (4, 52) = 161.52; $p < 0.001$; $\eta^2 p = 0.93$) and a significant "warm-up × recovery" interaction (F (8, 104) = 12.24; $p < 0.001$; $\eta^2 p = 0.48$) (Table 2). Figure 7 shows that after 10 min and 20 min of stopping anaerobic exercise, following a 15-min warm-up, the TBARSs concentrations are significantly lower than those recorded after 5- or 10-min warm-ups ($p < 0.05$).

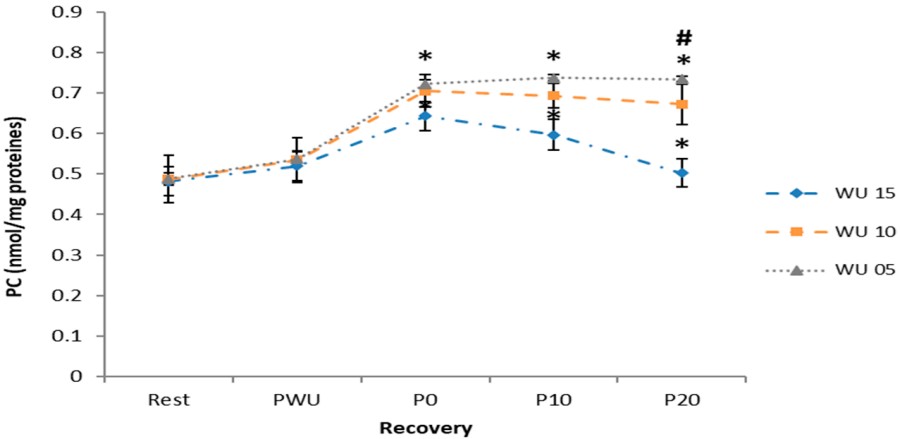

**Figure 7.** Evolution of PC at recovery after the three warm-up durations. *: significant difference compared to 15-min warm-up at $p < 0.05$; #: significant difference compared to 10-min warm-up at $p < 0.05$.

PC concentrations continue to increase until 20 min of recovery; except under the effect of the 15-min warm-up, there was a significant drop and a decreasing trend under the effect of the 10-min warm-up, but without statistical significance (Table 2). ANOVA showed that the duration of the warm-up, recovery, and the "warm-up × recovery" interaction has significant effects on PC concentrations at $p < 0.001$ after the Wingate test. PC concentrations were significantly lower under the effect of the 15-min warm-up (Figure 8).

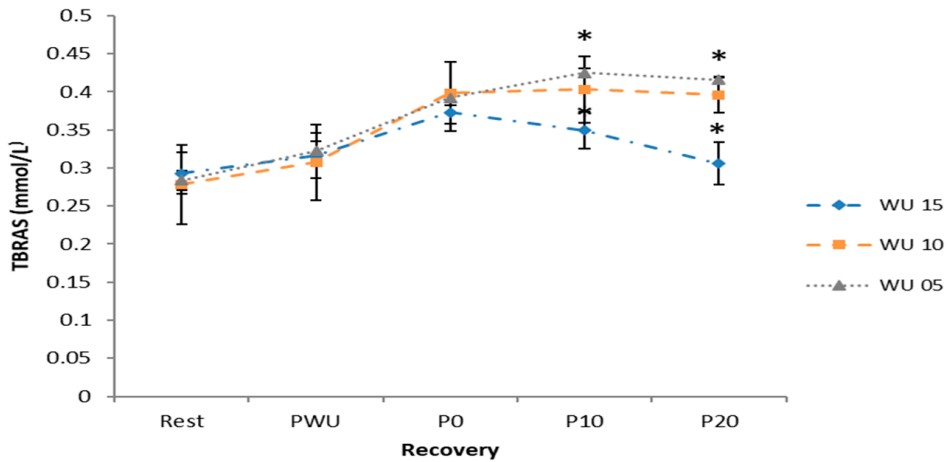

**Figure 8.** Evolution of TBARSs at recovery after the three warm-up durations. *: significant difference compared to 15-min warm-up at $p < 0.05$.

## 4. Discussion

Warming up, in addition to reducing the risk of injury [22,25,26], attempts to increase physical performance in order to prepare players for competitions in the best possible condition [27]. As a result, a suitable warm-up regimen should be of sufficient duration to prevent tiredness. The purpose of this study was to investigate whether three different warm-up lengths affected markers of oxidative stress after exercise and recovery after anaerobic exercise in amateur handball players.

This study shows that a 15-min warm-up after 20 min of recovery following anaerobic exercise increases recovery and the return of plasma antioxidants to their baseline levels. Positive changes in all enzymatic antioxidants evaluated accompany the improvements.

In our investigation, anaerobic exercise followed by a warm-up was related to an increase in antioxidant plasma concentrations. The three warm-up times resulted in significant increases in SAT, GPx, MDA, TBARSs, and PC in all subjects. Our results followed similar trends to several other studies that have found an increase in plasma MDA and SAT concentrations, as well as blood concentrations of GPx, SOD, and GR, immediately following a 30 s Wingate test in trained judokas [32]. An increase in lipid peroxidation (MDA) occurred, following sprints, strength exercises, and the Wingate test [33–35] as well as increased plasma concentrations of MDA, SOD, GPX, and GR after the 30 s Wingate test in young adults [36]. The increase in SOD was statistically significant after the two warm-up protocols (5 min and 10 min). On the other hand, a downward trend is observed, during recovery, under the effect of the 15-min warm-up, although the difference is not significant. Nevertheless, our findings are in contrast to those of Groussard et al. [9] in whom a Wingate test causes a decrease in SOD activity without modification of GPX activity. This divergence could be explained, in part, by the diversity of the three protocols carried out, the level of physical aptitude which differs between the subjects as well as the individualized response to exercise which could bring exaggerated reactions of oxidative stress during recovery.

The same finding, concerning blood GR concentrations, was observed experimentally, under the effect of warming up for 10 min and 15 min. However, after 20 min from the end of the exercise, this parameter's concentrations find their initial values under the effect of the 15-min warm-up with statistically significant differences from those recorded under the effect of 5-min and 10-min warm-ups. The present study showed that a Wingate test, preceded by a three-duration warm-up protocol (5 min, 10 min, or 15 min), resulted in an increase in SAT and MDA concentrations up to a 20-min recovery. In addition, following the 15-min warm-up protocol and after 20 min of recovery, the decrease in SAT concentrations was statistically significant, although the difference is not significant compared to the other two warm-up times.

The 5-min and 10-min warm-up protocols showed a significant effect on GPx kinetics during recovery. Indeed, its concentrations continued to increase until 20 min of recovery. In contrast, the 15-min warm-up resulted in a partial drop in GPx concentrations from the cessation of anaerobic exercise until 20 min of recovery, although it was not significant relative to P0. The difference in concentrations at P20 after the 15-min warm-up compared to the other two warm-up protocols (5 min and 10 min), was statistically significant ($36.97 \pm 1.62$; $40.87 \pm 3.75$ and $41.55 \pm 2.55$, respectively).

A number of explanations could be presented to explain the wide range of responses. The first is connected to the duration of the warm-up, which varies from protocol to protocol. An inadequate warm-up period appears to impair physical performance and thus have varied impacts on oxidative stress during recovery. The second explanation would be related to the subjects' varying physical abilities. The differences in fitness levels between participants, as well as their individual responses to exercise, may result in increased oxidative stress reactions after recovery. Conditioned athletes will almost certainly require a longer warm-up to attain an appropriate rise in body temperature that results in a significant increase in muscle fiber conduction velocity [37,38]. Changes in antioxidant concentrations were shown at the end of the three warm-up procedures administered to our individuals. These are identified by significantly lower TBARSs and PC concentrations during recovery and after a 15-min warm-up compared to those reported after 5 or 10 min of warm-up. The longest heating time would be associated with the reduction in concentrations of these two measures to their baseline value during 20 min of recovery.

It is well accepted that there is a relationship between muscle damage and increased free radicals, which leads to oxidative stress [39]. The study by Maughan et al. [40], which discovered a delayed rise in lipid peroxidation produced in plasma during eccentric running exercise, has been proposed as evidence for the involvement of reactive oxygen species (ROS) in the muscle injury process. Furthermore, a relationship has been shown between muscle damage and the extent of neutrophil immigration, which can result in the generation of excess ROS during inflammatory processes [41]. In our study, the results obtained after the 15-min warm-up show the tendency of the concentrations of antioxidants to return to their baseline values. This can be explained by the beneficial effects of warm-up, which allows for an increase in temperature; this in turn improves blood flow to active tissues and facilitates oxygen dissociation from hemoglobin [6], an increase in the rate of nerve transition [42,43], and an increase in the speed and strength of muscle contraction. However, the lack of assessment of the concentrations of several non-enzymatic antioxidants, on the other hand, may be regarded as a limitation of the current investigation. Furthermore, future studies should focus on improving the warm-up program (length and intensity) in conjunction with stretching. Similarly, the small sample size may have hindered our capacity to identify variations in the parameters we chose. This is one of the work's limitations, and it should be considered in light of our findings.

## 5. Conclusions

According to our data, the two warm-up regimes (5 min and 10 min) were not associated with positive changes in antioxidant capacity during recovery in any of our participants. A 15-min warm-up, on the other hand, may help to control and mitigate the deleterious effects of oxidative stress, potentially minimizing muscle damage after anaerobic activity. This 15-min warm-up regimen appears to increase antioxidant levels, reducing oxidative stress during recovery.

*Practical Applications*

The current study demonstrates that athletes may benefit from a 15-min warm-up routine, which might positively modulate and reduce the detrimental effects of oxidative stress, perhaps reducing muscle damage and improving physical performance. It is there-

fore beneficial to add a 15-min warm-up program to the training and competitive session preparation regimens.

**Author Contributions:** Conceptualization, A.K. and M.E.; methodology, S.K.; writing—original draft preparation, A.K. and M.E.; writing—review and editing, A.K., S.K. and M.E. All authors have read and agreed to the published version of the manuscript.

**Funding:** This research received no external funding.

**Institutional Review Board Statement:** The experimental protocol was approved by the Research Ethics Committee of the Faculty of Medicine, University of Sfax, Tunisia (IRB00009853-03).

**Informed Consent Statement:** Informed consent was obtained from all subjects involved in the study.

**Data Availability Statement:** Not applicable.

**Acknowledgments:** The authors wish to express their sincere gratitude to all the participants who took part in this study for their maximal effort and cooperation. The author M. Elloumi would like to thank Prince Sultan University for the support and for paying for the APC.

**Conflicts of Interest:** The authors have no conflict of interest that are directly relevant to the content of this article.

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
