# Peer review of "Effect of Different Warm-Up Durations on the Plasma Oxidative Stress Biomarkers Following Anaerobic Exercise in Amateur Handball Players"

_applsci, doi:10.3390/app131910576_

Round 1

Reviewer 1 Report

The research is very interesting.

The reviewer suggests conducting research for professional athletes in order to compare the results.

Author Response

Responses to Reviewer 1

Manuscript ID: applsci-2537483, entitled "Effect of different warm-up durations on the plasma oxidative stress biomarkers following anaerobic exercise in amateur handball players.”

Dear Reviewer 1,

The authors would like to thank you for the time you have allocated to the expertise of our paper, with accurate, pertinent and constructive remarks and suggestions. Please find below our responses to your valuable comments.

Reviewer 2 Report

Please write the hypotheses of the study   The criteria for inclusion and exclusion of the participants from the study should be written in detail.   Please add the sample size calculation to the method section.   Please detail the dietary program section. What kind of a standardized program was applied to the participants? explain   Why were times such as 5 minutes, 10 minutes and 15 minutes used for warm-up times? explain based on scientific source..   How 3 different warm-up times were applied to 14 people.. explain in detail.. You can put a table..   Who analyzed the blood samples from the participants.. How the blood samples were transported and analyzed on the bruise..Detail the blood sampling section..   line 83-85 please base it on scientific source..   Please elaborate on the section on anthropometric measurements. Have the ACSM criteria been observed? Also, there is no explanation in the method section on how to determine FFM, LBM in Table 1. Please add   Table 2 contains French words. Please correct it.   Calculate and add the effect sizes for pairwise comparisons between names and protocols.   Please improve the limitations section..   Add the practical applications section

Moderate English revision is required

Author Response

(The authors gave the same response as above.)

Reviewer 3 Report

Dear authors,

I have sent you my opinions. I don't know if they will thank you or not, but they are sincere and I hope they will help you in your work to improve your studies.

Observation 1

You talk about the warm-up and its effects, but the handball players' warm-up does not take place in the specific position of the Wingate test. The specific movements of the test are not at all similar to the means used by handball players in the warm-up. The test blocks the upper limbs and explosive strength movements involving anaerobic effort specific to the upper body are not performed.

It would probably have been enough for young people, non-athletes or other subjects. It could very well be other categories. Why did you choose handball, because I don't see the connection with handball!!! Please justify the choice made.

Observation 2

The Wingate test, by using the bicycle / cycle ergometer, eliminates or diminishes the presence of gravity, a fact that is not found in specific handball movements. Please argue this aspect to me from your point of view.

Observation 3

The introduction is slightly short, too simple and does not sufficiently argue, through studies, the oxidative effects of exercises generated by heating. The introduction does not refer to those parts of the warm-up that would involve anaerobic exercise and the relationship to accidents or the induction of oxidative stress. Because not all means in a sports warm-up can so easily cause burns or injuries.

Observation 4

2.1. Participants: I would like to know what handball experience the 14 subjects had at the time of testing? How many years of handball experience did they have? Or at least how many years of consistent exercise? No data are known about their possible adaptations to exercise (bradycardia, lactate tolerance, etc.). We are presented with age, weight, height and mass index. OK and so. But when we talk about athletes, selected, even amateurs, these aspects must be highlighted, because they make the difference in tests, especially on oxidative stress as a result of anaerobic effort.

How do the authors see this aspect? Please correctly and completely detail the data about the participants. What does "amateur handball players" mean in the case of the subjects in this study? (number of training sessions / week)????

2.2. Experimental design: In the two weeks between tests, did the subjects practice handball? Were they in the recovery or transition period between matches, stages, etc??? Specify this aspect.

Observation 5

Table no. 2, the table header must be in English, not in French

Observation 6

The figures should be clearer, better quality

Observation 7

L219-221 repeats the same ideas as in the introduction (L33-L45). Please remove or bring other ideas with arguments in favor of what you want to say. Don't repeat the same idea, in other words.

Observation 8

L228-230 is the purpose of the study, which has no place in the discussion here. You have already passed it once, at L65-L68.

Please remove L216 to L230: these are not discussions of what you have discovered, discussions of the results. They are ideas that are to be passed in the introduction, but you have already included them there.

Observation 9

L231 please reword ”Our results are consistent with”, because the subjects of those studies were trained sportsmen, not amateurs (as your subjects are) Maybe your results are similar / close / etc etc……because you still bring it up ”the level of physical aptitude which differs between the subjects” ?????

Observation 10

L265 : ”The second explanation would be related to the subjects' varying physical abilities.”  - there was no condition for inclusion or exclusion of subjects in the study group. You are missing such conditions from 2.1.. Maybe you should rearrange, specifying much better the criteria for excluding and including subjects.

Observation 11

L266-268 :

Conditioned athletes will almost cer-266 tainly require a longer warm-up to attain an appropriate rise in body temperature that 267 results in a significant increase in muscle fiber conduction velocity.” – is it your results or a reference to another study. If it is the result of other studies, then specify the source!!!

Observation 12

L275-276 ” It is well accepted that there is a relationship between muscle damage and increased free radicals, which leads to oxidative stress [39].– specify how it relates to your research? Because as far as I understand, this is not what you are aiming for.

Observation 13

”ROS” -  I first found this formulation at L278. What does it represent?

Observation 14

I do not see the direct connection of the ideas written by you in the discussions, at L275-279, with your study and measurements and results !!??

Observation 15

I think the limits of this research are many more than just one (of the small number of subjects). Please reflect on this aspect and specify more, because there are many more limits.

Observation 16

in order to remedy the known adverse effects of oxidative stress on muscle damage after anaerobic exercises” – ” muscle damage”???? I do not believe in any way that this conclusion has anything to do with your research procedure. It is not the purpose of this research.

Observation 17

Discussions need to be better constructed, starting from what the authors have determined and comparing or making analogies with what is already studied or determined by other authors. I recommend the authors of this study to eliminate unfounded personal opinions, if they are not based on their own research, or on the research of other authors (in which case they should be cited).

Observation 18

The basic idea is ok, but its way of presentation and argumentation should be much improved, for clarity, ease of study and understanding.

The authors did not invent an extraordinary product. And a simple product can be edifying, but it must be presented properly. Success

Check certain abbreviations of data to be in English and the head of table no. 2 (it is in French). Thank you

Author Response

Responses to Reviewer 3

Manuscript ID: applsci-2537483, entitled "Effect of different warm-up durations on the plasma oxidative stress biomarkers following anaerobic exercise in amateur handball players.”

Dear Reviewer 3,

The authors would like to thank you for the time you have allocated to the expertise of our paper, with accurate, pertinent and constructive remarks and suggestions. Please find below our responses to your valuable comments.

Round 2

Reviewer 2 Report

All corrections were made.. It is accept in current form..

Reviewer 3 Report

Dear authors,

Your manuscript is better, after the first part of the reviews, but for some of my observations you avoided arguing with me and I don't feel that you responded to my requests (you cannot describe a category of subjects, very small by the way, with a single added sentence and so little data about them).

I personally am not clear about the aspects that make the connection in a warm-up, the tests on a static support (which does not resemble the effort time in handball) and the body's response to effort in handball (as a variable of oxidative stress on which you followed). Surely you also know the fact that the time of effort, the form in which it takes place, the position in which it takes place clearly and directly influences the type of residues and the body's reactions). You can't tell me that a bike test is equivalent to a specific effort in orthostatism. But let's say that it was more difficult to find another more appropriate tool.

Honestly, for some of my observations you evaded the answer and did not argue.

We, as reviewers, only express our opinion to correct, help or facilitate the expression of an idea. But that idea must be real. scientifically supported and to help the practice of the field.

From my point of view, the article should be filtered and argued better.

I will leave it to the editor's decision to choose further.

success